 | 

# Transposons are important contributors to gene expression variability under selection in rice populations

**Raúl Castanera[1]\*, Noemia Morales-Díaz[1], Sonal Gupta[2], Michael Purugganan[2,3], Josep M Casacuberta[1]\***

[1]Centre for Research in Agricultural Genomics, CRAG (CSIC-IRTA-UAB-UB), Campus UAB, Cerdanyola del Vallès, Barcelona, Spain; [2]Center for Genomics and Systems Biology, New York University, New York, United States; [3]Center for Genomics and Systems Biology, New York University Abu Dhabi, Saadiyat Island, Abu Dhabi, United Arab Emirates

**\*For correspondence:**
raul.castanera@cragenomica.
es (RC);
josep.casacuberta@cragenomica.
es (JMC)

**Competing interest:** The authors declare that no competing interests exist.

**Abstract** Transposable elements (TEs) are an important source of genome variability. Here, we analyze their contribution to gene expression variability in rice by performing a TE insertion polymorphism expression quantitative trait locus mapping using expression data from 208 varieties from the *Oryza sativa* ssp. *indica* and *O. sativa* ssp. *japonica* subspecies. Our data show that TE insertions are associated with changes of expression of many genes known to be targets of rice domestication and breeding. An important fraction of these insertions were already present in the rice wild ancestors, and have been differentially selected in indica and japonica rice populations. Taken together, our results show that small changes of expression in signal transduction genes induced by TE insertions accompany the domestication and adaptation of rice populations.

## eLife assessment

This **valuable** study reports on the role of transposable elements in gene expression variation in rice and how TE-associated expression changes could have been selected during domestication. The combination of evidence from linkage studies and selection scans for a subset of insertions is **convincing**, although it is difficult to know in how many cases linkage of TE insertions to other regulatory variants is responsible for altered gene expression and in how many cases the TE insertions themselves are the bona fide cause of altered gene expression. The work will be of interest to colleagues working on the role of transposable elements in adaptation and to biologists working on domestication.

## Introduction

Transposable elements (TEs) are a major component of eukaryotic genomes, particularly in plants (*Wendel et al., 2016*). Their ability to move, creating mutations by insertion/excision, and amplify in genomes – thus generating new copies that provide opportunities for recombination – make them a major source of genetic variability (*Tenaillon et al., 2010*). For this reason, TEs are considered a major driver of plant genome evolution, both in the wild and under human selection (*Lisch, 2013*). TE insertions and other structural variants, however, have been frequently overlooked when using genome-wide association studies (GWAS) to look for genetic variants linked to interesting phenotypes (*Voichek and Weigel, 2020*). The recent development of efficient tools to identify transposon insertion polymorphisms (TIPs) (*Vendrell-Mir et al., 2019*) has allowed the incorporation of TIPs in

these analyses, and several GWAS reports have shown that TIPs uncover additional variability linked to phenotypic traits (*Akakpo et al., 2020*; *Domínguez et al., 2020*; *Castanera et al., 2021*). Interestingly, TIPs often explain more phenotypic variance compared to single nucleotide polymorphisms (SNPs), and can be used for genomic prediction (*Vourlaki et al., 2022*). The reason for this could be that, as compared with SNPs, TE insertions are more frequently the causal mutation.

A major fraction of the mutations linked to crop domestication and breeding are associated with changes of the expression of genes involved in signal transduction (*Swinnen et al., 2016*; *Meyer and Purugganan, 2013*). TEs can alter gene transcription, activating or repressing them, by different means. On the one hand, insertion of TEs within a promoter can interfere with transcription, and the silencing of TEs inserted close to genes can result in repression of gene expression (*Wang et al., 2020*; *Martin et al., 2009*; *Quadrana, 2020*). Alternatively, TE insertions can also result in gene overexpression (*Tian et al., 2022*; *Studer et al., 2011*), as TEs bring their own transcriptional promoters that can induce expression of neighboring genes. This is particularly clear for LTR retrotransposons which contain promoters in their 5' and 3' LTRs, and may provide nearby genes with alternative promoters. In addition, TEs can also contain transcription factor-binding sites (TFBS) that can alter the expression of host genes (*Rebollo et al., 2012*); it has been shown, for example, that miniature inverted-repeat transposable elements (MITEs) have frequently amplified and redistributed TFBS in plant genomes (*Morata et al., 2018*).

Here, we explore the impact of the movement of TEs during the recent evolution of rice on the variability of gene expression in this species. Rice is one of the most important food crops in the world, and varieties are generally found in two subspecies – *Oryza sativa* ssp. *japonica* and ssp. *indica*. Other minor variety groups include aus varieties, which appear closely related to indica, and basmati rice varieties that appear to be a hybrid between ssp. *japonica* and aus. We took advantage of a recently published transcriptional analysis of 208 rice varieties belonging to the major rice variety groups grown under wet paddy and intermittent drought conditions (*Groen et al., 2020*) to perform an expression quantitative trait locus (eQTL) GWAS using TIPs as genetic markers. We show that TIPs are frequently associated with changes of expression of rice genes and that many TIPs altering the expression of regulatory genes have been positively selected in indica and japonica rice subspecies.

## Results

### TIPs are associated with gene expression variation in rice

We used a recently published dataset containing genome resequencing and transcription data for rice (*O. sativa*) varieties to perform TIP-eQTL mapping analyses. The data we analyzed consist of an 'indica' dataset for 126 *O. sativa* ssp. *indica* and some *circum*-aus accessions, and 82 'japonica' accessions comprising of *O. sativa* ssp. *japonica* and some *circum*-basmati varieties, described in *Groen et al., 2020*. We predicted TIPs using PopoolationTE2 (*Kofler et al., 2016*), and identified 45,050 TIPs. Using the Minghui MH63 short-read data and assembled genome (*Zhang et al., 2016*), we estimated the performance of the TIP genotyping on this dataset to have 97.1% sensitivity and 92.2% precision (see methods). Most of the TIPs detected are DNA transposons (84%), especially MITEs belonging to the *Stowaway* (24%) and *Tourist* (20%) superfamilies; LTR retrotransposons make up a minor fraction (9 %).

The 45,050 TIPs are distributed along the 12 chromosomes and 81.3% of them are less than 5 kb away from an annotated gene (IRGSP RAP-DB gene models). This results in up to 81.4% of rice genes having a TIP less than 5 kb distant. In order to analyze the potential of TIPs to generate gene expression variability in rice, we used the 3' mRNA-seq expression data from leaves of adult plants grown in normally watered soil ('wet' condition) and under intermittent drought stress ('drought stress' condition) (*Groen et al., 2020*). We selected data from 15,549 genes showing expression in more than 10% of the samples. Due to the strong population structure present in rice we separately analyzed data in the indica (126 varieties) and japonica (82 varieties) datasets. We found a total of 563 significant associations between TIPs and gene expression levels in the indica population and 356 in the japonica population (TIP-eQTLs with minor allele frequency (MAF) >3%, false discovery rate [FDR] adjusted p-value <0.05 in at least one replicate) (*Figure 1A, B*, *Supplementary file 1*). These involve 477 and 317 genes, in indica and japonica, respectively, representing 3.1% and 2% of the expressed genes. The TIP-eQTLs were related to the different TE types (orders *Wicker et al., 2007*), and the proportion

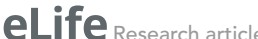

**Figure 1.** Characterization of rice transposon insertion polymorphism (TIP)-expression quantitative trait loci (eQTLs). (**A, B**) Association between TIPs and gene expression levels in cis (TIP-eQTLs) found in indica and japonica population along the 12 rice chromosomes. Horizontal lines represent 5% significance thresholds corrected for multiple testing using false discovery rate (FDR) method. (**C**) Number of TIP-eQTLs found in each condition classified at the order transposable element (TE) level. White bars represent the number of total TIPs (secondary axis). (**D**) Expression variance explained by leading TIP-eQTL and SNP-eQTL. Each point represents a gene. Indica and japonica TIP-eQTLs are combined into a single plot. Number of TIP-eQTLs (**E**) or SNP-eQTL (**F**) per Mb of gene feature. Upstream and downstream regions are 1 kb long. Gene region includes 5' and 3' UTRs. All the TIPs and SNPs with a significant association (FDR <5%) were used in this analysis.

The online version of this article includes the following figure supplement(s) for figure 1:

**Figure supplement 1.** Expression variance explained by leading transposon insertion polymorphism (TIP)-expression quantitative trait locus (eQTL) and SNP-eQTL in the associations detected for each gene (dot).

of TE orders found in the significant TIPs did not differ significantly from their representation across the genome (*Figure 1C*).

In order to estimate the relative contribution of TIPs and SNPs to gene expression changes, we also performed eQTL mapping analyses using a high density genotype matrix consisting of 824,311 SNPs for indica and 706,542 SNPs for japonica (*Groen et al., 2020*). We obtained 4913 SNP-eQTLs in indica and 3308 SNP-eQTLs in japonica associated with SNPs located less than 5 kb from the gene

(were each SNP-eQTL represents the leading SNP–gene association). This corresponds to ~31.6% of the genes in indica and ~21.3% in japonica. For each gene with a significant TIP-eQTL association, we compared the proportion of variance ($R^2$) explained by the most significantly associated TIPs and/or SNPs. For most genes that have an associated TIP, either there is no associated SNP or the TIP explains more expression variance than the associated SNP (~62% of the indica TIP-eQTLs and ~73% of the japonica TIP-eQTLs) (*Figure 1D*). This result could be reproduced by running the association using SNPs and TIPs together (*Figure 1—figure supplement 1A*), and also by subsampling SNPs to the same number of TIPs (*Figure 1—figure supplement 1B*). This demonstrates that TIPs uncover genetic associations with changes of expression that are not observed when using SNPs as molecular markers.

The SNP-eQTLs found in our study are associated to both positive (47%) and negative (53%) effects on gene expression levels, very close to an even distribution. In contrast, TIPs more frequently have a negative effect (59%). The proportion varies among TE superfamilies, with TIR/MULEs being close to 50%, and LTR/Gypsy having the most frequently negative impact on gene expression (67%, *Supplementary file 2*). This suggests that, as compared with SNP-eQTLs, TIP-eQTLs are more frequently the likely causal mutation responsible for the change in expression, and that the effect of the insertion is more generally negative, in line with recent analyses done in *Capsella* (*Uzunović et al., 2019*), especially for long elements such as LTR retrotransposons.

An analysis of the TIP- and SNP-eQTL location with respect to the associated genes shows that while SNP-eQTLs are more evenly distributed across the genic region (*Figure 1F*), TIP-eQTLs are over-represented in the 1 kb region upstream of the gene (*Figure 1E*), which is the region that most frequently contains promoter elements regulating transcription. This suggests that although some TIP-eQTLs could be in linkage disequilibrium with the causal mutation for the change of expression, an important fraction of TIP-eQTLs could be the actual causal mutation inducing expression variation.

It has been proposed that an important fraction of gene expression variation is deleterious, and therefore, alleles associated with major expression effects should be maintained at lower frequencies (*Josephs et al., 2015*; *Lye et al., 2022*). An analysis of the effect size of the different TIP-eQTLs taking into account their frequency in the population shows that, indeed, low- and high-frequency TIPs (corresponding to rare alleles) show the highest positive or negative effect on the expression of the associated gene (*Figure 2A–D*, *Figure 2—figure supplement 1*). This is consistent with what is described as the rare variant effect (*Lye et al., 2022*) where rare (and likely deleterious) mutations in a population are associated with greater effects on gene expression in a population.

A comparison of TIP-eQTLs associated with changes of expression in the two different growth conditions (wet and drought stress) shows that there is an important overlap of eQTLs between the two environments (38.5% for indica TIP-eQTLs and 33.2% for japonica TIP-eQTLs) (*Figure 2E*). Such overlap increases to 90% (average of indica and japonica, wet) and 71% (average of indica and japonica, drought stress) when we compare TIP-eQTLs of one condition with non-FDR-corrected TIP-eQTLs in the other environment (*Figure 2F*). All the associations shared in the two environments had the same effect type on expression (positive or negative). This result suggests that the majority of TIP-eQTLs are general associations with changes of expression in both growth conditions, and that only a small number of TIP-eQTLs (81 in indica and 78 in japonica) are associated with stress-specific changes of expression. Nevertheless, among the TIP-eQTLs associated with changes of gene expression in both wet and drought stress, there are TIP associated with genes known to be involved in drought tolerance (*Supplementary file 1*). As an example, TIP_49046 is associated with a reduction of the expression of the gene synaptotagmin-5 (*OsSYT-5*). *OsSYT-5* encodes a $Ca^{2+}$ sensing protein with a C2 domain, is expressed in both stressed and non-stressed plants, and it has been recently shown that its silencing enhances drought tolerance in rice (*Shanmugam et al., 2021*). Other examples are TIP_23764 and TIP_52367, associated with an increased expression of the gene encoding the OsSAPK10 ABA-activated protein kinase and the S-type euonymus-related lectin gene OsEULS2 in, respectively, two genes known to mediate drought stress in rice (*Gao et al., 2022*; *Lambin et al., 2020*).



**Figure 2.** Effect size and population frequencies of wet and drought-stress transposon insertion polymorphism (TIP)-expression quantitative trait loci (eQTLs). Representation of the effect size (beta) of the TIP-eQTLs with respect to their population frequency in indica (**A, C**) and japonica (**B, D**) for positive (**A, B**) and negative (**C, D**) effects. Venn diagram illustrating the intersection between TIP-eQTLs detected in each condition of the two

*Figure 2 continued on next page*

*Figure 2 continued*

subspecies analyzed, indica (ind) and japonica (jap) (**E**). Percentage of condition-specific (**F**) and species-specific (**G**) TIP-eQTLs. Shared TIPs are those falling in the intersection between the false discovery rate (FDR)-corrected TIP-eQTLs found in a given population/condition and all the associations of the other population/condition using a relaxed cutoff (p < 0.05, no FDR correction). Relationship between the frequencies in indica and japonica populations of the and 829 TIP-eQTLs (**H**) and the 33,389 non-eQTL-TIPs (**I**).

The online version of this article includes the following figure supplement(s) for figure 2:

**Figure supplement 1.** Effect size of transposon insertion polymorphism (TIP)-expression quantitative trait loci (eQTLs) based on their population frequency on the drought condition.

**Figure supplement 2.** Population frequencies of transposon insertion polymorphism (TIP)-expression quantitative trait loci (eQTLs) and TIP-no-eQTLs present at 1 kb upstream genes.

## TIP-eQTLs are present at different population frequencies in indica and japonica

Although the overlap of TIPs associated with changes in expression under wet and drought stress conditions is very high, the overlap between indica and japonica TIP-eQTLs is low, and only 90 TIP-eQTLs out of the 563 from indica and 356 from japonica are significantly associated with variation of gene expression in both subspecies (*Figure 2E*). Even when we make the less-stringent comparison of the TIP-eQTLs of one subspecies with the non-FDR-corrected TIP-eQTLs of the other, as much as ~69% of indica and ~60% of japonica TIP-eQTLs appear as subspecies specific (*Figure 2F*). Regarding the common associations between subspecies, all of them have the same effect type on expression (positive or negative). This result may suggest differences in the transcriptional networks in indica and japonica or, alternatively, that different alleles showing expression level differences may be differentially represented in the two subspecies. An analysis of the frequencies in indica and japonica of all the TIP-eQTLs described here shows that approximately one-third of the TIPs are only found in one of the two subspecies at MAF >3%, although only 59 (indica) and 21 (japonica) associations correspond to truly species-specific TIPs (completely absent in one of the two subspecies). We looked for these TIPs in 82 *Oryza rufipogon* and *Oryza nivara* varieties and found that 56% of the indica-specific and 24% of the japonica-specific TIPs were present in such population. This suggests that TEs have been actively inserting (or have been excised or eliminated from the population) very recently during rice evolution, likely post-domestication and during the crop diversification phase, as has already been proposed for some rice TE families (*Lu et al., 2017*; *Carpentier et al., 2019*). Here, we show that some of these recent TIPs are associated with changes in gene expression and may therefore have phenotypic consequences. Some of these insertions are found at high frequencies in its corresponding subspecies, which suggest that they may have been under positive selection (*Supplementary file 3*).

However, the vast majority of TIP-eQTLs found in this study (90%) correspond to TIPs present in both subspecies, which suggests that these insertions are relatively old and were already present in the ancestor of indica and japonica rice. Remarkably, in most cases they are found at very different frequencies in the two subspecies (*Figure 2H*). This suggests that these insertions were already present in the ancestor of indica and japonica rice and have been differentially retained in the two subspecies. Interestingly, an analysis of all non-eQTL-TIPs shows that an important fraction is found at the same frequency in both genomes (*Figure 2I*). This differential pattern between eQTL and non-eQTL-TIPs is also found when we consider only the upstream gene regions (1 kb, *Figure 2—figure supplement 2*), suggesting that the preferential retention of TIP-eQTLs in indica or japonica may be a specific phenomenon potentially linked to their impact on the expression of the associated genes.

We examined whether TIP-eQTLs present in both indica and japonica varieties are also found in either *Oryza rufipogon* and *Oryza nivara*, the wild rice relatives believed to be the ancestors of domesticated rice (*Choi et al., 2017*). We looked for the presence of the TIPs identified in rice in a set of 72 accessions of *O. rufigogon* and 10 accessions of *O. nivara* (collectively called rufipogon/nivara from now on) (*Supplementary file 4*). Up to 552 of the 829 TIP-eQTLs present in indica and/or japonica (66%), can be found in rufipogon/nivara varieties, confirming that an important fraction of TIP-eQTLs found in indica and japonica may be relatively old insertions already present in the wild ancestors of domesticated rice. Not surprisingly, in most cases the TIP frequencies in rufipogon/nivara are different from the frequencies of these insertions in indica and/or japonica (*Supplementary file 5*); these may

have arisen either due to selection or genetic drift accompanying the bottleneck associated with crop evolution.

In order to identify possible selection of TIPs in indica and japonica populations, we used the Population Branch Statistic (PBS) method (*Yi et al., 2010*) which examines strong differentiation in frequencies between populations. This approach consists in comparing the three pairwise $F_{ST}$ values between indica, japonica, and rufipogon/nivara to estimate the frequency change in TIPs that occurred since the divergence of the two rice subspecies from its wild ancestor. We calculated PBS values for a total of 11,698 TIPs present in the japonica, indica, and rufipogon/nivara populations, which included 354 TIPs that are eQTLs in in indica and/or japonica populations. TIP-eQTLs showed much higher PBS values in comparison to those not identified as eQTLs (*Figure 3A*) (mean absolute PBS of 0.06 vs 0.02, respectively, p < 0.01, Wilcoxon test). Furthermore, TIPs with the most extreme PBS values (above 95 percentile) are enriched for TIP-eQTLs (Fisher's exact test odds ratio = 4.87, p < 0.01), suggesting that a high fraction of TIP-eQTLs have been positively selected in indica or japonica. Note that some of the TIPs not identified as eQTLs in this particular dataset may actually be linked to variation of gene expression in organs, developmental stages or environmental conditions different from the ones analyzed here. Interestingly, up to 35% of the TIP-eQTLs identified here, associated with 119 genes, fell within the top 10% of absolute PBS values for all TIPs (*Figure 3A*), suggesting that TIPs associated with gene expression variation have likely been under differential selection during rice evolution.

We observe that TIPs can show evidence of selection in japonica, indica or both subspecies. As examples, *Figure 3B, C* shows the representation of the PBS analysis, and the frequency in the populations, of two TIP-eQTL whose frequency has greatly increased in indica (TIP_53500) or japonica (TIP_72732), compared with the mean PBS for all TIPs.

## Gene variants selected during rice evolution: some examples

Genes linked with TIP-eQTLs showing extreme indica or japonica PBS metrics are good candidates for genes underlying adaptation between different rice subspecies. Among the genes with TIP-eQTLs that have extreme PBS values in indica or japonica we can find several examples that are known to regulate plant architecture, plant and grain development or abiotic stress responses.

Some of the most extreme PBS values are for TIPs associated with changes of expression of genes involved in the signal transduction of hormones, including brassinosteroids, ABA, ethylene, jasmonic acid (JA) and auxin (*Supplementary file 5*). We found four different TIP-eQTLs with high PBS values associated with the EG2/OsJAZ1 gene (Os04g0653000), a locus encoding a JA signaling repressor (*Cai et al., 2014*); the four insertions are related to MITEs of the Tourist transposon superfamily. Three of the insertions (TIP_32891, TIP_32892, and TIP_32894), located upstream (3.4 and 2.7 kb) and downstream (150 bp) of the gene, are associated with an increase of expression of *EG2*, and are physically linked to one another (mean $r^2$ value = 0.88). In contrast, the fourth insertion (TIP_32893), which is not linked to them, is present within the first intron of the gene and is associated with expression reduction.

The recent advances in characterizing the pangenome of rice and the super-pangenome that includes its wild ancestors (*Qin et al., 2021*; *Zhou et al., 2020*; *Shang et al., 2022*) has allowed us to characterize the locus in indica, japonica, and rufipogon/nivara. This analysis showed that *EG2* is primarily found in two different haplotypes (*Figure 4A*). One haplotype, Hap A, contains the three MITE insertions associated with higher expression of EG2 (*Figure 4A, B*, and *Figure 4—figure supplement 1*), while the second haplotype, Hap B, contains the fourth MITE insertion and is associated with reduced EG2 expression (*Figure 4A, B* and *Figure 4—figure supplement 1*). Hap A was identified to be at high frequency in rufipogon/nivara (48%), whereas Hap B is at lower frequency (16%). The frequency of both Hap A and Hap B is slightly increased in japonica (65% and 28%, respectively). In indica, however, there is a clear reduction in the frequency of Hap A and a concomitant strong frequency increase of Hap B, associated with reduced expression in indica (16% and 83%, respectively). Interestingly, EG2/OsJAZ1 is a repressor of spikelet development (*Cai et al., 2014*) and the number of differentiated spikelets per panicle tends to be lower in japonica as compared with indica (*Ansari et al., 2003*).

We looked more closely for signs of selection at this specific locus, by examining levels of nucleotide diversity (π; *Nei and Li, 1979*) and diversity in haplotypes homozygosity tracts (H12; *Garud and Rosenberg, 2015*). Consistent with the expression (and possible spikelet phenotypic differences), we



**Figure 3.** Signatures of positive selection on transposon insertion polymorphism (TIP)-expression quantitative trait loci (eQTLs). (**A**) Absolute Population Branch Statistic (PBS) of 354 TIP-eQTLs (left) and 11,344 TIPs (no-eQTL, right) present in indica, japonica, and rufipogon populations. Dotted vertical lines represent the 95th and 99th percentiles of the PBS values of the whole dataset (11,698 TIPs). Red dots represent two examples of TIP-eQTLs with extreme PBS values. (**B**) Population frequency of the two TIPs with extreme PBS values, marked as red dots in the left panel (**A**), as well as of the whole TIP dataset. (**C**) Fst-based tree of the two TIPs with extreme PBS values, as well as of the whole TIP dataset (average Fst).

**Figure 4.** Selection on transposon insertion polymorphism (TIP)-expression quantitative trait loci (eQTLs) associated with *EG2* expression. (**A**) Representation of the two main *EG2* haplotypes present in rice and rufipogon/nivara populations identified in the rice super-pangenome. Conserved nucleotide regions are connected by gray marks. Structural variants longer than 50 bp are shown as white spaces. TIP-eQTLs are shown as red boxes.

*Figure 4 continued on next page*

*Figure 4 continued*

Additional TIPs are shown as white boxes. (**B**) Boxplot representation of the expression of the two *EG2* haplotypes in the indica population. Numbers inside boxplots represent the number of accessions in each group. Analysis of nucleotide diversity (π) and diversity in haplotypes homozygosity tracts (H12) for Hap A (**C, E**) and Hap B (**D, F**) in japonica (**C, D**) and indica (**E, F**) populations. The vertical dotted black line shows the position of the TIP insertion. The EG2 gene is schematically shown in blue. The horizontal dashed black line represents the mean of 1000 random permutation pulls (for p-value see *Supplementary file 6*).

The online version of this article includes the following figure supplement(s) for figure 4:

**Figure supplement 1.** Boxplots representing the expression differences between haplotypes in OsJAZ1.

found that in japonica the haplotype associated with increased *EG2* expression (Hap A) was under positive selection, whereas the haplotype associated with decreased expression of *EG2* (Hap B) did not show any sign of selection; this indicates that positive selection acts on individuals carrying the haplotype for high expression of *EG2* in japonica (*Figure 4C, D*). Specifically, we observe significantly lower π and higher H12 in Hap A, associated with increased *EG2* expression, while this pattern was not significant for Hap B (*Supplementary file 6*). However, in indica the opposite pattern appears to prevail, wherein there is a stronger evidence of selection for Hap B, but less so for Hap A (*Figure 4E, F*). This suggests positive selection toward higher expression of *EG2* in japonica, and lower expression of *EG2* in indica, which could potentially explain the higher number of spikelets per panicle observed in the latter (*Ansari et al., 2003*).

Another good candidate for genes underlying adaptation of rice is the *OsGAP* gene. There are two different TIP-eQTLs with high PBS values associated with changes of expression of the *OsGAP* gene (Os07g0500300) in japonica, whereas they do not correlate with changes in expression in indica (*Figure 5—figure supplement 1*). *OsGAP* encodes a putative GTPase activating protein, similar to CAR proteins (C2-domain abscisic acid-related proteins), that play an important role in ABA signal transduction in *Arabidopsis* (*Xu et al., 2019*). The first TIP (TIP_50057) is located ~4 kb upstream of the OsGAP gene and is associated with an increase of expression of the gene in japonica whereas the second insertion (TIP_50059) is located within the first intron and is associated with a decrease in expression in japonica (*Figure 5A, B*). The analysis of the locus using the super-pangenome data shows that *OsGAP* is found in two different haplotypes, one containing TIP_50057 (Hap A) and the other containing TIP_50059, together with some additional structural differences (Hap B; *Figure 5A*). The two TIPs defining the two haplotypes are present in the rufipogon/nivara population at complementary frequencies (73% and 19%, respectively). Interestingly, the proportion of the two haplotypes in cultivated rice seem to be very different, with an increased frequency of the haplotype associated with a reduced expression of *OsGAP* (Hap B) in both japonica (~50%) and indica (~95%) were it reaches near fixation.

It has been proposed that *OsGAP* is a negative regulator of ABA signaling in seed germination and dormancy, and that reduced expression of *OsGAP* may prevent rice pre-harvest sprouting (PHS) (*Xu et al., 2019*). Reduced seed dormancy is a common target for selection of cultivated varieties, but this trait may come at a cost of a higher PHS risk, which may be a problem, especially in regions where heavy rain is common during the harvest season. Therefore, the appropriate degree of dormancy may depend on the agroecological conditions in which a particular variety is grown. We see here that two independent TIPs are associated with changes of *OsGAP* expression in japonica but not in indica (*Figure 5A*, *Figure 5—figure supplement 1*). Interestingly, there are clear and strong signs of selection for increased expression of *OsGAP* in japonica, wherein Hap A was under positive selection, and Hap B was selectively neutral, indicating positive selection acts on only those individuals carrying haplotype for high expression of *OsGAP* in japonica (*Figure 5C, D*; *Supplementary file 6*). In contrast, there is no evidence for selection in indica varieties despite the near fixation of one haplotype. This suggests that an increased level of *OsGAP* may be relevant for the control of dormancy in japonica but not in indica (despite its near-fixation in the latter), which would be in line with recent data suggesting that seed dormancy is regulated by different genes/alleles in indica and japonica (*Magwa et al., 2016*).

Finally, we find a TIP-eQTL whose frequency has greatly increased in cultivated rice with respect to rufipogon/nivara. TIP_45706, corresponding to an insertion of a gypsy-like LTR-RT at ~1 kb upstream of the *OsMPH1* gene (*Figure 6A*). This insertion is associated with a decrease in the expression of

**Figure 5.** Selection on transposon insertion polymorphism (TIP)-expression quantitative trait loci (eQTLs) associated with *OsGAP* expression.
(**A**) Representation of the two main *OsGAP* haplotypes present in rice and rufipogon/nivara populations identified in the rice super-pangenome.
Conserved nucleotide regions are connected by gray marks. Structural variants longer than 50 bp are shown as white spaces. TIP-eQTLs are shown
as red boxes. Additional TIPs are shown as white boxes. (**B**) Boxplot representation of the expression of the two *OsGAP* haplotypes in the japonica
population. Numbers inside boxplots represent the number of accessions in each group. Analysis of nucleotide diversity (π) and diversity in haplotypes
homozygosity tracts (H12) for Hap A (**C**) and Hap B (**D**) in japonica population. The vertical dotted black line shows the position of the TIP insertion. The
*OsGAP* gene is schematically shown in blue. The horizontal dashed black line represents the mean of 1000 random permutation pulls (for p-value see
*Supplementary file 6*).

The online version of this article includes the following figure supplement(s) for figure 5:

**Figure supplement 1.** Boxplots representing the expression differences between haplotypes in OsGAP.

*OsMPH1* in indica and possibly japonica varieties (*Figure 6B*, *Figure 6—figure supplement 1*), and
is found at low frequency in rufipogon/nivara (~11%) but at high frequency in both indica (70%) and
japonica (95%). *OsMPH1* encodes an MYB-like transcription factor that has been shown to regulate
plant height, its reduced expression resulting in shorter plants (*Zhang et al., 2017*). A reduction in
plant height due to a mutation in the *SD1* gene, which encodes the gibberellin biosynthesis gene



**Figure 6.** Selection on transposon insertion polymorphism (TIP)-expression quantitative trait loci (eQTLs) associated with *OsMPH1* expression. Representation of the two main *OsMPH1* haplotypes present in rice and rufipogon/nivara populations identified in the rice super-pangenome. Conserved nucleotide regions are connected by gray marks. Structural variants longer than 50 bp are shown as white spaces. TIP-eQTLs are shown as red boxes. (**B**) Boxplot representation of the expression of the two *OsMPH1* haplotypes in the japonica population. Numbers inside boxplots represent the number of accessions in each group. Analysis of nucleotide diversity (π) and diversity in haplotypes homozygosity tracts (H12) for Hap A in indica population (**C**) and in japonica population (**D**). The vertical dotted black line shows the position of the TIP insertion. The OsMPH1 gene, which is schematically shown in blue. The horizontal dashed black line represents the mean of 1000 random permutation pulls (for p-value see ***Supplementary file 6***).

The online version of this article includes the following figure supplement(s) for figure 6:

**Figure supplement 1.** Boxplots representing the expression differences between haplotypes in OsMPH1.

GA-20ox, was at the origin of the Green Revolution in the 1960s, but it has been shown that alleles of *SD1* resulting in shorter culm length were also selected during the domestication of japonica rice (*Asano et al., 2011*). Our results suggest possible parallel selection of alleles in other genes that may lead to similar phenotypes. This is further reinforced by strong selection acting on the haplotype containing the TIP_45706 insertion in both indica and japonica (*Figure 6C, D*, *Supplementary file 6*), leading to reduced expression of *OsMPH1* and thus shorter plants.

## Discussion

Transposons are a major source of genome variability, and are known to affect gene expression in numerous ways. The importance of rare variants and unfixed TE insertions for gene expression variation in humans (*Goubert et al., 2020*) and plants (*Uzunović et al., 2019*) has already been recently described. A recent pangenome analysis in tomato has shown that the phenotypic variation of important crop traits is linked to structural variants present in the species, which are associated to subtle changes of transcription of genes involved in signal transduction. Many of these SV-eQTLs are likely TIPs, and some lead to phenotypic consequences such as the jointless trait caused by a transposon insertion, which allows complete separation of fruits from other floral parts (*Alonge et al., 2020*). In the present study we evaluated the importance of TIP-related expression variability in the recent evolution of rice. To this end we performed a TIP-eQTL mapping using expression data from rice varieties from the *O. sativa* ssp. *indica* and *O. sativa* ssp. *japonica* subspecies. We show here that using TIPs in addition to SNPs as genetic information allows the uncovering of additional genomic associations to changes in gene expression, and that when both TIPs and SNPs are both associated, TIPs often explain more of the variance in expression. This is in line with recent reports that have also used TIPs for GWAS with different phenotypes in tomato and rice (*Domínguez et al., 2020*; *Castanera et al., 2021*), and suggests that, even if some of the TIPs described here are probably just linked to the causal mutation of the phenotype (i.e., an SNP or a different type of SV), they may be more frequently the causal mutation themselves as compared with SNPs. The concentration of TIP-eQTLs in upstream regions of genes, where most transcriptional regulatory elements usually reside, also suggest that a high fraction of the TIP-eQTLs here described are the actual mutation underlying gene expression variation.

The close association of some TEs with genes could also partially explain the frequent association of more than one TIP with changes of expression of particular genes. Indeed, among the 718 genes with TIP-eQTLs described here, 18% have two or more TIPs associated with changes of expression. Interestingly, among the 30 genes with TIPs in indica and japonica that have the highest PBS – an indicator of strong differential selection – 43% have more than one TIP-eQTL. This suggests that TIPs are an important source of gene expression variability in rice, in particular in genes that may have been strongly selected during its recent evolution. In some cases, the different TIPs linked to a gene are associated with opposite effects on its expression and are present in different haplotypes. In these instances, as shown for the *OsJAZ1* and *OsGAP* genes, the two haplotypes have strong signs of selection, although one positively and the other negatively, which suggests that both TE insertions may have played a role in the diversification of rice subspecies.

The results presented here show that most TIPs associated with variation of gene expression in indica and/or japonica are relatively old insertions that are also present in rufipogon/nivara, the wild ancestors of rice. This is in line with recent data showing that a high number of the structural variants associated with changes of expression in tomato were already present in its wild ancestor (*Alonge et al., 2020*) and highlights the importance of the standing variation, already present in the wild ancestors, for crop adaptation. It is assumed that most TE insertions are selectively neutral or slightly deleterious (*Arkhipova, 2018*), and the presence of TIPs associated with expression variation of genes in the wild ancestors of rice, suggests that this variation can be well tolerated in wild rice. Interestingly, we show here that many of these TIPs have been positive or negative selected in rice populations, which shows that they translate into selectable phenotypic differences in the agroecological conditions of cultivated rice. Indeed, we show examples of selected variants with modified expression of genes known to be linked to important traits that were targets of selection. It has been already proposed that a major fraction of the mutations linked to crop domestication and breeding are associated with changes of gene expression involved in signal transduction (*Swinnen et al., 2016*; *Meyer and Purugganan, 2013*). Here, we show that specific expression variants of genes involved in

signal transduction have been differentially selected in indica and japonica rice populations. In addition, our results also point to TEs as a major driver of gene expression variation selected during crop adaptation and breeding.

## Methods

### TIP detection

Resequencing data for 126 indica and 82 japonica rice accessions were obtained from *Groen et al., 2020* (Bioproject accessions PRJNA557122, PRJNA422249, and PRJEB6180). BBDuK (https://sourceforge.net/projects/bbmap/) was used for adaptor and quality trimming. Clean reads were aligned to the Nipponbare reference genome (*Matsumoto et al., 2005*) using BWA (*Li and Durbin, 2010*). PoPoolationTE2 (*Kofler et al., 2016*) was used to detect TIPs in the mode 'joint' using Nipponbare TE annotation described by *Ou et al., 2019*. TIPs with a zygosity lower than 0.25 in all samples were excluded to avoid false positives. TIPs were further filtered using the parameters --min-count 5, --max-otherte-count 2, --max-structvar-count 2, and only those having MAF higher than 3% and no missing data were kept. Finally, the TIP matrix was transformed to binary form using zygosity cutoff of 0.05 to define an insertion as present.

### Evaluation of TIP-calling performance in our dataset

Resequencing data from 48 indica rice accessions randomly sampled from our dataset were used to identify TIPs with the objective of evaluating the performance of TIP calling. MH63 was included among these accessions. This variety has a chromosome-level assembly as well as short-read resequencing data publicly available. MH63 sequencing short reads were obtained from NCBI SRA accession SRX1639978 and subsampled to 15× to match the mean coverage of the full dataset. TIP detection was carried out using Nipponbare genome as reference, and the 'joint' mode of PopoolationTE2, with the parameters described in the Methods section. We used RepeatMasker to annotate TEs in these MH63 assembled regions, and used the annotated TEs to benchmark precision and sensitivity of the TIP calls, according to the following formulas:

$$Sensitivity = TP/(TP + FN)$$
$$Precision = TP/(TP + FP)$$

True positives (TP) were TIPs detected by PopoolationTE2 that could be detected in the MH63 orthologous region by RepeatMasker. False positives were TIPs detected by PopoolationTE2 that had no TE of the same family annotated by RepeatMasker in the corresponding region of MH63. False negatives (FN) were cases when the TIP prediction of MH63 was an absence, but the assembled region contained the TE detected by RepeatMasker.

### TIP- and SNP-eQTL mapping

Transcriptome data for the 208 rice accessions (wet and drought stress conditions; three independent replicates per condition) were obtained from *Groen et al., 2020* and separated into individual replicate matrices. Transcripts expressed in more than 10% of the samples on each replicate were extracted and counts were normalized using the *vst* function of DESeq2 (*Love et al., 2014*). TIP- and SNP-eQTL mapping were performed using Matrix eQTL software (*Shabalin, 2012*), applying the simple linear regression model and including subpopulation groups as covariates. We used a cutoff distance of 5000 bp to identify *cis*-eQTLs and a 5% FDR threshold for multiple testing corrections.

### Selection analyses

For the PBS analysis, we looked for the presence of rice TIPs in the 82 accessions belonging to the rufipogon/nivara population (*Supplementary file 4*) and retained only those present in the three populations (indica, japonica, and rufipogon/nivara), resulting in a matrix of 13,622 TIPs and 288 accessions. The TIP matrix was filtered to remove TIPs with more than 5% missing data, or with MAF <5%. The remaining missing data (1.7%) were imputed using the 'wright' algorithm implemented in SNPready. The clean matrix (11,698) was used to calculate the individual PBS values per TIP, following the formulas described in *Yi et al., 2010*.

Along with PBS, to evaluate whether TIP-eQTLs show signs of selection, we estimated nucleotide diversity (π; *Nei and Li, 1979*), which is a site-frequency spectrum measure to test for the presence of selection. This was done using a custom script (*Gupta, 2023*) in a window of 25 SNPs with a sliding window of 5. Further, we also estimated the H12 homozygosity statistic (*Garud and Rosenberg, 2015*) that can detect the presence of both hard and soft sweeps associated with selection. Since we expect long runs of homozygosity in the regions around the selected loci, H12 was calculated in windows of 50 SNPs, with a sliding window of 25. Both these statistics were estimated separately for indica and japonica, and for the samples with TIP insertions present and absent. This was for a range of 100 kb (50 kb up- and downstream the TIP insertion), using SNPs identified from *Groen et al., 2020* but without the 1000 bp linkage thinning. To test for the significance of the statistics, we performed a permutation test (*N* = 1000) with random sampling with the same number of individuals that were used to estimate the selection statistics. The statistics were estimated for these 1000 random pulls in the same regions, and using the same window sizes as the original statistics.

## Acknowledgements

We thank Jae Yong Choi, Simon C Groen, and Sebastián Ramos-Onsins for helpful discussions.

## Additional information

### Funding

| Funder | Grant reference number | Author |
| --- | --- | --- |
| Ministerio de Ciencia e Innovación | PhD Student Fellowship. PRE2020-095111 | Noemia Morales-Díaz |
| Ministerio de Ciencia e Innovación | Postdoctoral Fellowship. IJC2020-045949-I | Raúl Castanera |
| Ministerio de Ciencia e Innovación | PID2019-106374RB-I00 | Josep M Casacuberta |
| Severo Ochoa Programme for Centres of Excellence in R&D | SEV-2015-0533 | Josep M Casacuberta |
| Ministerio de Ciencia e Innovación and CERCA Program / Generalitat de Catalunya | CEX2019-000902-S | Josep M Casacuberta |
| National Science Foundation | IOS-1546218 | Michael Purugganan |
| National Science Foundation | IOS-2204374 | Michael Purugganan |
| Zegar Family Foundation | | Michael Purugganan |
| NYU Abu Dhabi Research Institute | | Michael Purugganan |

The funders had no role in study design, data collection, and interpretation, or the decision to submit the work for publication.

### Author contributions

Raúl Castanera, Conceptualization, Software, Formal analysis, Investigation, Visualization, Methodology, Writing – original draft, Writing – review and editing; Noemia Morales-Díaz, Formal analysis, Methodology, Writing – review and editing; Sonal Gupta, Formal analysis, Methodology, Writing – original draft, Writing – review and editing; Michael Purugganan, Writing – original draft, Writing – review and editing; Josep M Casacuberta, Conceptualization, Funding acquisition, Investigation, Writing – original draft, Writing – review and editing

## Author ORCIDs
Raúl Castanera http://orcid.org/0000-0002-3772-7727

Reviewer #1 (Public Review): https://doi.org/10.7554/eLife.86324.3.sa1
Reviewer #2 (Public Review): https://doi.org/10.7554/eLife.86324.3.sa2
Author Response https://doi.org/10.7554/eLife.86324.3.sa3

# Additional files

## Supplementary files

• Supplementary file 1. Extended information on significant transposon insertion polymorphism (TIP)-expression quantitative trait loci (eQTLs).

• Supplementary file 2. Transposon insertion polymorphism (TIP)-expression quantitative trait locus (eQTL) effect on gene expression levels up, for increased gene expression when tansposable element (TE) is present, down for decreased gene expression when TE is present according to TE superfamily classification.

• Supplementary file 3. Extended information on indica- and japonica-specific transposon insertion polymorphisms (TIPs) associated with expression changes.

• Supplementary file 4. NCBI-SRA accession numbers for *O. rufipogon* and *O. nivara* accessions used in this study.

• Supplementary file 5. Population Branch Statistic (PBS) results and transposon insertion polymorphism (TIP) information for TIP-expression quantitative trait loci (eQTLs) present in indica, japonica, and rufipogon/nivara populations.

• Supplementary file 6. Statistical significance of the permutation test performed to identify selection on OsJAZ1, OsGAP, and OsMPH1. Only the haplotypes that cleared the p-value cutoff of 0.01 were reported to be under selection.

• MDAR checklist

## Data availability

Resequencing data are available at SRA Bioproject accessions PRJNA557122, PRJNA422249 and PRJEB6180. Original expression dataset is available at Zenodo (doi: https://doi.org/10.5281/zenodo.3533431). Filtered expression dataset, TIP and SNP matrices and code to reproduce the analyses are available at Zenodo (doi:https://doi.org/10.5281/zenodo.7646220) and github (https://github.com/gsonal802/TIPeQTL_Selection_Osativa, *Gupta, 2023*).

The following dataset was generated:

| Author(s) | Year | Dataset title | Dataset URL | Database and Identifier |
|---|---|---|---|---|
| Castanera R | 2022 | Transposons are a major contributor to gene expression variability under selection in rice populations | https://zenodo.org/record/7646220 | Zenodo, 10.5281/zenodo.7646220 |

The following previously published datasets were used:

| Author(s) | Year | Dataset title | Dataset URL | Database and Identifier |
|---|---|---|---|---|
| Groen et al. | 2019 | Processed RNA expression count data from Groen et al.: The strength and pattern of natural selection on rice gene expression | https://zenodo.org/record/3533431#.Y-4w9RPMKpo | Zenodo, 10.5281/zenodo.3533431 |

*Continued on next page*

*Continued*

| Author(s) | Year | Dataset title | Dataset URL | Database and Identifier |
|---|---|---|---|---|
| Groen et al. | 2019 | Population genome sequencing of Asian rice Oryza sativa varities | https://www.ncbi.nlm.nih.gov/bioproject/?term=PRJNA557122 | NCBI BioProject, PRJNA557122 |
| Groen et al. | 2019 | Oryza sativa (Asian cultivated rice) | https://www.ncbi.nlm.nih.gov/bioproject/?term=PRJNA422249 | NCBI BioProject, PRJNA422249 |
| Groen et al. | 2019 | Rice3k | https://www.ncbi.nlm.nih.gov/bioproject/?term=PRJEB6180 | NCBI BioProject, PRJEB6180 |

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
