## [Editor Report · eLife assessment]

This **valuable** study reports on the role of transposable elements in gene expression variation in rice and how TE-associated expression changes could have been selected during domestication. The combination of evidence from linkage studies and selection scans for a subset of insertions is **convincing**, although it is difficult to know in how many cases linkage of TE insertions to other regulatory variants is responsible for altered gene expression and in how many cases the TE insertions themselves are the bona fide cause of altered gene expression. The work will be of interest to colleagues working on the role of transposable elements in adaptation and to biologists working on domestication.

---

## [Referee Report · Reviewer #1 (Public Review)]

For many years it has been understood that transposable elements (TEs) are an important source of natural variation. This is because, in addition to simple knockouts of genes, TEs carry regulatory sequences that can, and sometimes do, affect the expression of genes near the TEs. However, because TEs can be difficult to map to reference genomes, they have generally not been used for trait mapping. Instead, single nucleotide polymorphisms are widely used because they are easy to detect when using short reads. However, improvements in sequencing technology, as well as an increased appreciation of the importance of TEs to both linked to favorable alleles and are more likely to be causing the changes that make those alleles beneficial in a given environment. Further, because TE activity can occur after bottlenecks, they can provide polymorphisms in the absence of variation in point mutations.

In this manuscript, the authors carefully examine insertion polymorphisms in rice and demonstrate linkage to differences in expression. To do this, they used expression quantitative trait locus (eQTL) GWAS using TIPs as genetic markers to examine variation in 208 rice accessions. Because they chose to focus on genes that were expressed in at least 10% of the accessions, presumably because more rare variants would end up lacking statistical power. This is an understandable decision, but it says that recent insertions, such as the MITE elements detailed in a previous paper, would not be included. Importantly, although TIPs associated with differentially expressed genes are far less common than SNPs' traditional eQTLs, there were a significant number of eQTLs that showed linkage to TIPs but not to QTL.

The authors then show that of the eQTLs associated with both TIPs and SNPs, TIPs are more tightly linked to the eQTL, and are more likely to be associated with a reduction in expression, with variation in the effects of various TEs families supporting that hypothesis. Here and throughout, however, the distance of the TEs could be an important variable. It is also worth noting the relative numbers in order to assess the claim in the title of the paper. The total number of eQTL-TIPs is ten-fold less than the number of eQTL-SNPs, and, of the eQTLs that have both, there are a significant number of eQTL-TIPs that are not more tightly linked to the expression differences than the eQTL.

The authors show that eQTL-TIPs are more likely to be in the promoter-proximal region, but this may be due to insertion bias, which is well documented in DNA-type elements. Here and throughout the authors are careful to state that the data is consistent with the hypothesis that TEs are the cause of the change, but do not claim that the data demonstrate that they are.

Throughout the rest of the manuscript, the authors systematically build the case for a causal role for TEs by showing, for instance, that eQTL-TIPs show much stronger evidence for selection, with increased expression being more likely to be selected than decreased expression. The authors provide examples of genes most likely to have been affected by TE insertions.

Overall, the authors build a convincing case for TEs being an important source of regulatory information. I don't have any issues with the analysis, but I am concerned about the sweeping claims made in the title. Once you get rid of eQTLs that could be altered by either SNPs or TIPs and include only those insertions that show strong evidence of selection, the number of genes is reduced to only 30. And even in those cases, the observed linkage is just that, not definitive evidence for the involvement of TEs. Although clearly beyond the scope of this analysis, transgenic constructs with the TEs present or removed, or even segregating families, would have been far more convincing.

The fact that many of the eQTL-TIPs were relatively old is interesting because it suggests that selection in domesticated rice was on pre-existing variation rather than new insertions. This may strengthen the argument because those older insertions are less likely to be purged due to negative effects on gene expression. Given that the sequence of these TEs is likely to have diverged from others in the same family, it would have been interesting to see if selection in favor of a regulatory function had caused these particular insertions to move away from more typical examples of the family.

---

## [Referee Report · Reviewer #2 (Public Review)]

In this manuscript, Castanera et al. investigated how transposable elements (TEs) altered gene expression in rice and how these changes were selected during the domestication of rice. Using GWAS, the authors found many TE polymorphisms in the proximity of genes to be correlated to distinct gene expression patterns between O. sativa ssp. japonica and O. sativa ssp. indica and between two different growing conditions (wet and drought). Thereby, the authors found some evidence of positive selection on some TE polymorphisms that could have contributed to the evolution of the different rice subspecies. These findings are underlined by some examples, which illustrate how changes in the expression of some specific genes could have been advantageous under different conditions. In this work, the authors manage to show that TEs should not be ignored when investigating the domestication of rise as they could have played an important role in contributing to the genetic diversity that was selected. However, this study stops short of identifying causations as the used method, GWAS, can only identify promising correlations. Nevertheless, this study contributes interesting insights into the role TEs played during the evolution of rice and will be of interest to a broader audience interested in the role TEs played during the evolution of plants in general.

---

## [Author Response]

The following is the authors' response to the original reviews.

**Reviewer #1 (Public Review):**
[…] Overall, the authors build a convincing case for TEs being an important source of regulatory information. I don't have any issues with the analysis, but I am concerned about the sweeping claims made in the title. Once you get rid of eQTLs that could be altered by either SNPs or TIPs and include only those insertions that show strong evidence of selection, the number of genes is reduced to only 30. And even in those cases, the observed linkage is just that, not definitive evidence for the involvement of TEs. Although clearly beyond the scope of this analysis, transgenic constructs with the TEs present or removed, or even segregating families, would have been far more convincing.

We notice that the referee thinks that we "built a convincing case for TEs being an important source of regulatory information". This is what we wanted to convey in the title, were we were cautious to not claiming that TEs are the most important contributor to gene expression variability in rice populations. However, we agree with the referee that the title may be improved to better describe the results presented. We have therefore changed the title to "Transposons are an important contributor to gene expression variability under selection in rice populations".

With respect to demonstrating causality by removing or introducing the TEs, this is indeed a work we plant to do but that, as stated by the referee, is beyond the scope of this analysis.

The fact that many of the eQTL-TIPs were relatively old is interesting because it suggests that selection in domesticated rice was on pre-existing variation rather than new insertions. This may strengthen the argument because those older insertions are less likely to be purged due to negative effects on gene expression. Given that the sequence of these TEs is likely to have diverged from others in the same family, it would have been interesting to see if selection in favor of a regulatory function had caused these particular insertions to move away from more typical examples of the family.

The TIP-eQTL are from different classes, superfamilies and families and the number of TIP-eQTLs of the same family is too small to deduce sequence communalities (4.6 TIP-eQTLs/family in indica and 3.6 TIP-eQTLs/family in japonica). On the other hand the effect of TIPs on expression can be positive or negative (we show actually that it is often negative). In the later case, a plausible scenario would be of the insertion inactivating a promoter element, and in this case it would be the insertion itself, and not the actual sequence of the TE what would be selected.

Also, previous work done in our lab has shown that TEs can amplify and mobilize transcription factor binding sites that are bound by the TF even when they are not close to a gene and therefore probably not directly affecting gene expression (Hénaff et al.,2014. The Plant Journal). In that case, the sequence of the eQTL TEs and those that are far away from genes will not necessarily differ.

**Reviewer #2 (Public Review):**
In this manuscript, Castanera et al. investigated how transposable elements (TEs) altered gene expression in rice and how these changes were selected during the domestication of rice. Using GWAS, the authors found many TE polymorphisms in the proximity of genes to be correlated to distinct gene expression patterns between O. sativa ssp. japonica and O. sativa ssp. indica and between two different growing conditions (wet and drought). Thereby, the authors found some evidence of positive selection on some TE polymorphisms that could have contributed to the evolution of the different rice subspecies. These findings are underlined by some examples, which illustrate how changes in the expression of some specific genes could have been advantageous under different conditions. In this work, the authors manage to show that TEs should not be ignored when investigating the domestication of rise as they could have played an important role in contributing to the genetic diversity that was selected. However, this study stops short of identifying causations as the used method, GWAS, can only identify promising correlations. Nevertheless, this study contributes interesting insights into the role TEs played during the evolution of rice and will be of interest to a broader audience interested in the role TEs played during the evolution of plants in general.

We agree with the referee that the results presented do not allow concluding on causality, and we have been careful not to pretend they would in the manuscript. We plan to perform analysis of adding or removing TEs by CRIPR/Cas 9 approaches to address this, but, in line with referee's 1 comment, we think this is beyond the scope of this analysis.

**Reviewer #1 (Recommendations For The Authors):**
Everything that I need to say is provided in the public portion of my review.
**Reviewer #2 (Recommendations For The Authors):**
Major concerns:1. The authors compare the proportion of the variance explained by the most significant TIP and SNP on the observed eQLTs associated with TIPs and SNPs. Thereby the authors conclude that TIPs explain more variance than SNPs. If I am not mistaken the GWAS was run separately for TIPs and SNPs, however, I am wondering if running the GWAS on the combined TIP and SNP dataset might be the better way to compare the variance explained by TIPs and SNPs on gene expression differences. It would be nice to see if these results also hold true if a TIP and SNP combined dataset is used as the most significant marker in a GWAS might not be the causal mutation but might just be linked to the causal mutation. Further in the TIP dataset, the number of markers is only 45k and in the SNP dataset, it is 1 000k, which could bias the GWAS toward finding markers that explain more of the variation in the dataset with fewer markers.

We addressed the reviewer concern by using two complementary approaches, whose results are described in the text (lines 119-121) and in the new Figure 1-figure supplement 1.

First, we addressed the concern regarding the independent GWAS for TIPs and SNPs vs a combined strategy. For this, we built new japonica/indica genotype matrices containing all TIP and SNP matrix together and ran eQTL mapping again. Using the same strategy (association + FDR adjust), we found 100% of the previous TIP-eQTLs and 99% of the previous SNP-eQTLs. We repeated the same analysis (proportion of expression variance), and the results were mostly the same (Figure 1-figure supplement 1A).

Second, we addressed the two concerns (combined genotypes and different amount of TIP and SNP markers) using a single approach. SNP matrices were LD pruned using a r2 = 0.9 and later subsampled to the exact number of TIPs (Indica = 30,396, Japonica = 25,168). We verified that these SNPs covered well the 12 rice chromosomes. SNP and TIP genotypes were later merged into a single matrix, and eQTL mapping was repeated for each of the subspecies and conditions using the same parameters as in the previous version of the manuscript. 100 % of the previously reported TIP-eQTL associations were found using this new approach. Nevertheless, we found a very important drop of sensitivity in the SNP-eQTLs (only 15-20% of the previous associations were detected), possibly due to the strong reduction in the number of SNPs (> 95 %), which results in much lower number of markers at < 5Kb from genes. We repeated the analysis of Figure 1D, and observed very similar results (Figure 1-figure supplement 1D). There is a very important number of TIP-eQTL associations that do not coincide with SNP-eQTLs, (74% in indica, 83% in japonica) indicating that TIP-eQTL mapping is complementary to SNP-eQTL mapping as it uncovers additional associations (note that in this case the overlap between TIP-eQTLs and SNP-eQTLs is lower than in the previous analysis due to the lower sensitivity of SNP-eQTL mapping using less markers). In the cases were both a TIP and a SNP coincide as eQTL, TIPs explained slightly more variance than SNPs in both indica and japonica (in 54% of the cases TIP variance > SNP variance).

2. Line 146 to 152: in this section, the authors describe overlaps between TIP-eQTLs in two different growth conditions, however, in the text it is not mentioned if the TIPs have the same effect on gene expression in the two conditions or if the gene expression is up-regulated in one condition but down-regulated in the other. This information would be interesting to have here, especially as the authors go on to say that only a small number of TIP-eQTLs are stress-specific. The same comment also goes for the eQTL overlap described on lines 167 to 170.

We checked the effect type (positive or negative) of TIP-eQTLs in both scenarios (associations shared between wet/dry conditions, and associations shared between subspecies). In both cases, 100 % of the shared TIP-eQTLs have the same effect type in the two conditions or subspecies. We have updated the text accordingly (Lines 55-157 and Lines 179-181)

3. Lines 192 to 196: the authors mention that the frequency of non-eQTL-TIPs was at the same frequency in indica and japonica, which is in contrast to eQTL-TIPs. However, on line 132 it is mentioned that eQTL-TIPs were overrepresented in 1 kb regions upstream of genes. Hence, is the pattern of the frequency of non-eQTL-TIPs being at the same frequency in indica and japonica also observed in the 1 kb regions upstream of genes and/or if the distribution of non-eQTL-TIPs is matched to one of the eQTL-TIPs? Or is this pattern driven by non-eQTL-TIPs far away from genes?

We checked the frequencies of TIPs at 1Kb upstream genes and found that the general pattern is maintained, with the frequencies of TIP no-eQTLs being more correlated than that of TIP-eQTLs. We have included this information (lines 204-206) an added a new supplementary file (Figure 2-figure supplement 2)

4. In the discussion, the authors could briefly discuss how linked selection affecting TIPs could contribute to the observed results. After reading the second example in the result section where one of the example TIPs (TIP_50059) is found on the Hap B which contains "some additional structural differences" (line 290), I was left wondering how much of the increase in TIP frequency can be attributed to genetic hitchhiking? And how much of the results could be caused by linked selection, especially when considering that structural variations are not included in the GWAS analyses.

We agree with the referee in that some of the TIP eQTLs here described might be not the actual cause of expression variability (ej, TIP linked with the causal mutation), although we cannot know the exact fraction. This is stated in several places of the results and discussion sections. However, the fact that TIPs tend to explain more variance than SNPs and that TIP eQTL, but not SNP eQTL, tend to concentrate in the upstream proximal region of genes where most transcription regulatory sequences are located (Figure 1), suggest that TIP eQTLs could be more frequently the causal than SNP eQTLs. We revised the text to ensure that we convey this message appropriately.

Minor comments:• Lines 80 to 83: the description of the rice phylogeny should be moved to the introduction.

Done (Lines 68-72)

• Line 177 to 186: It was unclear to me if the authors checked in the ancestral rice population laced the TIPs described in this section as recently inserted in the indica and japonica ssp. It would be nice to add this information to this section.

Thanks to the referee comment we noted an imprecision in the text. The approximate 1/3 of subspecies specific TIP-eQTLs refers to the TIPs at 3% MAF (ie, some of these insertions could be present at > 3% in indica, but at < 3% MAF in japonica). We now indicate only the TIPs that are truly specific to any of the two subspecies (frequency is zero in one of the two) and looked for their presence in rufipogon:

59 insertions are indica-specific. Of those, 33 are present in rufipogon.

21 insertions are japonica-specific. Of those, 5 are present in rufipogon.

We have incorporated this information in the manuscript (Lines 185-189). The species-specific TIPs are also available in the Supplementary File 3.

• Line 353: "have two of more TIPs" should be "two or more"

Done (Line 369)

• Figure 1D: Using a square layout instead of a rectangle layout for the plot will make it easier to interpret.

Done.